# Pathophysiology and Diagnosis of ALS: Insights from Advances in Neurophysiological Techniques

**DOI:** 10.3390/ijms20112818

**Published:** 2019-06-10

**Authors:** Mehdi A. J. van den Bos, Nimeshan Geevasinga, Mana Higashihara, Parvathi Menon, Steve Vucic

**Affiliations:** Western Clinical School, University of Sydney, Sydney 2145, Australia; mehdivandenbos@gmail.com (M.A.J.v.d.B.); nimesh84@hotmail.com (N.G.); mana_higashihara@hotmail.com (M.H.); parmenon2010@gmail.com (P.M.)

**Keywords:** ALS, cortical hyperexcitability, glutamate excitotoxicity, TMS

## Abstract

Amyotrophic lateral sclerosis (ALS) is a rapidly progressive and fatal neurodegenerative disorder of the motor neurons, characterized by focal onset of muscle weakness and incessant disease progression. While the presence of concomitant upper and lower motor neuron signs has been recognized as a pathognomonic feature of ALS, the pathogenic importance of upper motor neuron dysfunction has only been recently described. Specifically, transcranial magnetic stimulation (TMS) techniques have established cortical hyperexcitability as an important pathogenic mechanism in ALS, correlating with neurodegeneration and disease spread. Separately, ALS exhibits a heterogeneous clinical phenotype that may lead to misdiagnosis, particularly in the early stages of the disease process. Cortical hyperexcitability was shown to be a robust diagnostic biomarker if ALS, reliably differentiating ALS from neuromuscular mimicking disorders. The present review will provide an overview of key advances in the understanding of ALS pathophysiology and diagnosis, focusing on the importance of cortical hyperexcitability and its relationship to advances in genetic and molecular processes implicated in ALS pathogenesis.

## 1. Introduction

Amyotrophic lateral sclerosis (ALS) is a rapidly progressive neurodegenerative disorder of the human motor system, clinically characterized by dysfunction of the upper and lower motor neurons, which forms the basis of diagnosis [1,2,3]. While disease onset is typically focal involving upper or lower limbs, bulbar or respiratory regions, the ensuing progressive course affects contiguous body regions resulting in global muscle weakness, with respiratory dysfunction representing a terminal phase of the disease [4,5]. Understanding the relationship between upper and lower motor neuron dysfunction is critical for unravelling ALS pathogenesis (Figure 1A), and three opposing theories have been proposed [6]. It has been suggested that ALS originates at a cortical level, with corticomotoneuronal hyperexcitability mediating neuronal degeneration via a transsynaptic anterograde mechanism, the dying forward hypothesis [7]. Others have proposed a contrasting theory, whereby lower motor neuron dysfunction was postulated to occur as a primary event, the dying back hypothesis [8,9]. A third school of thought, the independent hypothesis, proposed an independent and random degeneration of upper and lower motor neuron degeneration with patterns of disease spread being contiguous and random, conforming to underlying neuroanatomical boundaries [10,11].

Resolution of the relationship between upper and lower motor neuron dysfunction, particularly the site of disease onset, is critical in the understanding of ALS pathogenesis. Advances in neurophysiological techniques for assessing upper motor neuron function, particularly the development of the threshold tracking transcranial magnetic stimulation (TMS) technique [12], have suggested the importance of cortical hyperexcitability in ALS pathogenesis [13]. Consequently, novel treatment approaches aimed at modulating cortical hyperexcitability, utilizing repurposed therapeutic agents, may lead to the development of effective treatment strategies for ALS. The identification of cortical hyperexcitability may also be used as a diagnostic biomarker, facilitating an earlier diagnosis of ALS and enabling recruitment into clinical trials at a stage in the disease process where therapeutic agents are likely to be most successful [13]. The present review will provide an overview of key advances in the understanding of ALS pathophysiology and diagnosis, focusing on the importance of cortical hyperexcitability.

## 2. Cortical Physiological Dysfunction and ALS Pathogenesis

A complex interaction between genetic and environmental factors appears to underlie the development of ALS, leading to dysfunction of critical molecular pathways and ultimately neurodegeneration (Figure 2). Modelling studies have suggested that exposure to six consecutive steps or factors is required for the development of ALS [14], with the propensity to develop ALS acquired at conception [15]. While the nature of these steps or factors remain to be fully elucidated, a combination of genetic, environmental, and epigenetic factors have been postulated. Novel neurophysiological techniques have established that cortical hyperexcitability may be an important pathophysiological step [3].

## 3. Clinical Insights

The identification of concomitant upper and lower motor neuron signs as a pathognomonic feature of ALS suggests the importance of cortical dysfunction in ALS pathogenesis [2]. One hundred or so years later, Eisen and colleagues proposed the “dying forward” hypothesis, which postulated that hyperexcitability of descending corticomotoneuronal tracts mediated motor neuron degeneration via an anterograde transsynaptic glutamate excitotoxic process [7]. A number of specific clinical observations supported this dying forward mechanism including: (i) Muscles with a sparse corticomotoneuronal supply (extraocular and sphincter muscles) were relatively spared in ALS; (ii) pure lower motor neuron phenotypes of ALS were infrequent [16]; (iii) absence of ALS in the animal world, a finding ascribed to a developmental differences in corticomotoneuronal projections onto the spinal motor neurons across species [6,17]; (iv) asymmetric and focal motor deficits evident in the early stages of ALS could be explained by the complex anatomical association of corticomotoneuronal-anterior horn cell synaptic relationships [17].

The observation of dissociated muscle wasting as a specific clinical feature of ALS has further underscored the importance of cortical dysfunction in ALS pathogenesis [18]. The split-hand phenomenon refers to preferential wasting of the thenar group of intrinsic hand muscles, namely abductor pollicis brevis (APB) and first dorsal interosseous (FDI), compared to the hypothenar muscles (Figure 3) [19]. This dissociated pattern of muscle atrophy is specific for ALS, and is not evident in ALS neuromuscular mimicking disorders such as Kennedys disease or autoimmune motor neuropathy [20,21]. Cortical hyperexcitability appears to underlie the development of the dissociated pattern of muscle wasting in ALS [22]. In addition, a split-hand plus phenomenon, referring to preferential weakness of the APB muscle when compared to the flexor pollicis brevis, was also reported as a specific clinical feature of ALS mediated by cortical hyperexcitability [23,24]. More recently, split leg (preferential weakness of posterior calf muscles) and split elbow (preferential weakness of the biceps brachii compared to the triceps muscle) were also reported as specific clinical features of ALS and related to cortical dysfunction [25,26].

Concordance between the site of disease onset and handedness further underscores the importance of cortical mechanisms in ALS pathogenesis [27]. The site of disease-onset was more likely to be evident in the dominant hand of upper limb-onset ALS, while there was no correlation between footedness and site of onset [27]. Given that differences in cortical excitability have been established between the dominant and non-dominant hemispheres [28], changes in neuronal connectivity and cortical excitability were proposed as underlying mechanisms. The asymmetry of ALS onset and patterns of spread have also been ascribed to limb dominance, with corticomotoneuronal processes exerting an important pathogenic role [29]. Importantly, cortical hyperexcitability was reported to be an important physiological mechanism in mediating patterns of disease evolution in ALS [30]. 

## 4. Pathogenic Insights from Advances in Neurophysiological (TMS) Techniques in ALS

Assessment of cortical excitability with single, paired, and triple-pulse TMS techniques have significantly enhanced the understanding of cortical dysfunction in ALS pathogenesis, resulting in the development of novel diagnostic approaches [31,32]. In humans, TMS activates the motor cortex at a depth of approximately 1.5 to 2.0 cm by placing a magnetic coil on the surface of the scalp [33]. The magnetic pulse induces a current in the brain, flowing from a posterior-anterior direction, and resulting in the activation of cortical output cells (Betz cells) and intracortical neuronal networks within the primary motor cortex (M1) [34,35]. A descending volley (composed of direct and indirect waves) travels from the motor cortex (M1) to the spinal motor neurons via the corticomotoneuronal tracts (Figure 1B), leading to activation of the spinal motor neurons. This activity is reflected in the motor evoked potential [MEP] (Figure 1B), which is a surrogate biomarker of cortical excitability [32]. In a clinical setting, assessment of cortical excitability in ALS may be evaluated by utilizing paired- and single-pulse TMS techniques.

Paired-pulse TMS techniques have been important for assessing cortical physiology in ALS patients [31,36,37,38]. A number of TMS biomarkers have emerged with implementation of the paired-pulse technique, including short interval intracortical inhibition (SICI), intracortical facilitation (ICF) and short interval intracortical facilitation (SICF), all providing important insights into ALS pathogenesis, diagnosis, and clinical research [31,38]. Short interval intracortical inhibition is the most robust of the these parameters [13], and is generated when a subthreshold conditioning stimulus precedes a suprathreshold test stimulus at pre-determined interstimulus intervals [12,39]. In the original technique, the strength of the conditioning and test stimuli were kept constant, and changes in MEP amplitude were used as outcome measures [39,40,41]. Consequently, with interstimulus interval (ISI) set between 1 to 5 ms, the test MEP response was smaller or inhibited. Increasing the ISI between 7 to 30 ms resulted in larger MEP amplitudes, a phenomenon termed ICF [42]. Given that the “constant stimulus” paired-pulse technique was potentially limited by marked variability in MEP amplitudes [43,44], a threshold tracking technique was developed whereby an MEP response was fixed and the test stimulus tracked the fixed MEP response (Figure 4A,B) [12,45]. Short interval intracortical inhibition was reflected by an increase in test stimulus intensity required to track the fixed MEP response, while ICF was represented by the converse (Figure 4C). A recent study has reported greater reliability of the threshold tracking technique when compared to the constant stimulus method [46].

Physiological studies have indicated that SICI is mediated by inhibitory gamma-Aminobutyric acid (GABA)-ergic intracortical circuits, located within the primary motor cortex, acting via GABA_A_ receptors [34,35]. In addition, a weaker effect of glutamatergic neurotransmission has also been established [47]. In ALS patients, reduction or absence of SICI has been reported in sporadic ALS cohorts (Figure 4C), indicating cortical hyperexcitability [48,49,50,51,52]. Reduction of SICI is an early feature of ALS, preceding the onset of lower motor neuron dysfunction, correlating with biomarkers of peripheral neurodegeneration [13,50,53]. SICI reduction has also been established in atypical ALS phenotypes, including the clinically pure lower motor neuron syndromes [54]. Furthermore, the dissociated patterns of muscle atrophy, including the split hand and split hand-plus signs [22,24], as well as patterns of disease spread [30], have also been linked to cortical hyperexcitability. Importantly, reduction of SICI was shown to be an adverse prognostic factor in ALS [55], underscoring the pathogenic importance of cortical hyperexcitability.

Separately, SICI is partially normalized by riluzole [56], an anti-glutamatergic agent exhibiting modest clinical effectiveness in ALS [57,58]. These modulating effects of riluzole lasted approximately three months [59], paralleling the clinical efficacy of the drug. The reasons for the limited biological effectiveness of riluzole could be explained by overexpression of the blood brain barrier located efflux pumps, which evolve during the course of ALS [60]. Consequently, threshold tracking TMS may be used in future studies to assess the biological effectiveness of compounds at an early stage of drug development thus as to avoid unnecessary and costly Phase 3 trials.

Cortical hyperexcitability was also shown to be an early and prominent feature in familial ALS, including phenotypes linked to mutations in the superoxide dismutase-1 [52], fused in sarcoma (FUS) [61] and *c9orf72* genes [62], and correlating with peripheral neurodegeneration in the SOD-1 genotypes [63]. Interestingly, asymptomatic mutation carriers exhibited normal cortical excitability [52,62], with reduction of SICI preceding the clinical development of familial ALS by months [52].

Findings in sporadic and familial ALS cohorts have supported the notion that ALS is mediated by a multistep process [14], with cortical hyperexcitability being a critical step. Furthermore, dysfunction of intracortical interneuronal circuits appears to be critical in ALS pathogenesis, a notion supported by studies in the TDP-43^A315T^ mouse model, whereby hyperactivity of excitatory cortical interneurons underlies the development of hyperexcitability in cortical output tracts [64]. From a therapeutic perspective, focal ablation of the excitatory interneuronal circuits may lead to a normalization of excitability and potentially exert neuroprotective effects.

Short interval intracortical facilitation, also generated by the paired-pulse TMS technique, whereby a conditioning stimulus is set to peri- and suprathreshold levels followed by a test stimulus set at threshold intensity [41,65]. Recently, the threshold tracking TMS technique was adapted to generate SICF, revealing two distinct peaks at ISIs of 1.5 and 3 ms [66]. While the physiological processes underlying SICF development remain undetermined, a cortical origin has been proposed with SICF probably reflecting activity of facilitatory cortical circuits [66,67,68]. A significant increase in SICF was reported in sporadic ALS patients accompanying the reduction in SICI [69]. An index of excitation, a novel neurophysiological biomarker of cortical excitability expressing SICF as a function of SICI, was shown to be increased in ALS patients, suggesting that overactivity of facilitatory circuits contributed to cortical hyperexcitability [70]. Importantly, there was a significant correlation between the index of excitation and functional disability, underscoring the pathogenic importance of facilitatory circuit overactivity in the development of clinical features of ALS.

Single pulse TMS has also provided important insights into the role of cortical hyperexcitability in ALS pathogenesis. The cortical silent period (CSP) is a distinct biomarker of cortical inhibition, mediated by long-latency inhibitory circuits acting via gamma-aminobutyric acid type B (GABA_B_) receptors [32,42]. The CSP may also influenced by the density of the corticomotoneuronal projections onto motor neurons, the extent of voluntary drive and neuromodulators such as dopamine [42,47]. As with SICI changes, a marked reduction or absence of CSP duration has been reported in both sporadic and familial ALS phenotypes, being most prominent in the early stages of ALS [49,50,51,52,62,71,72,73,74,75,76,77,78,79]. Furthermore, the reduction in CSP duration has also been documented in atypical ALS phenotypes, such as the flail arm and leg variants of ALS [54,80]. This reduction of CSP duration represents degeneration and dysfunction of GABAergic inhibitory neurotransmission. As with SICI, the reduction of CSP duration is a specific feature of ALS in the context of neuromuscular diseases [13,51,74,81], further supporting the importance of cortical hyperexcitability in ALS pathogenesis.

Of further relevance, abnormalities in *motor thresholds* have been also reported in ALS [31]. The motor thresholds reflect the density of corticomotoneuronal projections onto the spinal and bulbar motor neuron, with thresholds being lowest in intrinsic hand muscles [82,83,84]. In addition, a variety of pharmacological agents modulate motor thresholds [36], with voltage-gated Na^+^ channel blocking agents increasing and enhancers of glutamatergic neurotransmission decreasing thresholds [85,86,87,88,89]. Reduction of motor thresholds (indicative of cortical hyperexcitability) has been reported in ALS, although normal thresholds have also been documented [30,49,50,52,90,91,92]. The reduction in motor thresholds is most prominent in the early stages of ALS, being associated with profuse fasciculations, preserved muscle bulk and hyper-reflexia [92,93], as well as contributing disease spread [30]. With disease progression, progressive increase in motor thresholds occurs, leading to inexcitability of the motor cortex [92], although this may not be an invariable finding [94]. Importantly, the change in motor thresholds was reported to be an independent predictor of cognitive dysfunction in the ALS phenotypes [95].

In conjunction with motor threshold changes, significant increases in MEP amplitudes have also been reported in ALS [31]. As with motor thresholds, the MEP amplitude reflects the density of corticomotoneuronal projections onto motor neurons [96] and is modulated by neurotransmitters such as glutamate [31,96]. Typically, the MEP response is expressed as a percentage of the maximum compound muscle action potential (CMAP) amplitude in order to eliminate a contribution of the peripheral nervous system, providing insights into the percentage of the activated corticomotoneuronal pool [38]. An increase in MEP amplitude has been established in sporadic and familial forms of ALS, being most prominent in early stages of the disease [13,22,50,52,53,54,62,76]. Similar to motor threshold findings, the increase in MEP amplitude is more prominent over the dominant motor cortex, contralateral to the side of disease onset, implying a role in disease progression [30]. The increase in MEP amplitudes correlate with surrogate biomarkers of motor neuronal degeneration, providing further support for the importance of cortical hyperexcitability in ALS pathogenesis [50,63], although a potential contribution from local spinal motor neuronal circuit hyperexcitability cannot be absolutely discounted [32].

An important issue to highlight is that some have suggested that cortical hyperexcitability is a compensatory response to spinal motoneuronal degeneration [49]. While this notion cannot be completely excluded, the absence of cortical hyperexcitability in ALS mimicking disorders, despite a comparable peripheral disease burden [13,51,74,81], would argue against a compensatory mechanism. In addition, molecular studies have provided additional credence for the importance of neuronal hyperexcitability in ALS [3]. At a molecular level, cortical hyperexcitability appears to be represented by glutamate excitotoxicity. Support for glutamate-mediated excitotoxicity was provided by significant reduction in expression and function of the astrocytic glutamate transporter (EAAT2) [97,98,99], which transports glutamate form the synaptic cleft into astrocytes thereby reduced glutamate activity [100]. Dysfunction of the EAAT2 transporter appears to occur pre-clinically in ALS [101,102], and activation of the EAAT2 transporter inhibitor, caspase-1, has been documented in the transgenic SOD-1 mouse model prior to onset of neuronal degeneration [101,102]. Conversely, increased expression of the EAAT2 transporter is neuroprotective in the SOD-1 model [103].

Morphological and functional cortical neuronal changes have been reported in pre-symptomatic SOD-1 mouse models [104,105,106,107,108], with neuronal hyperexcitability shown to be an early feature [104,109], lending further credence for the importance of cortical hyperexcitability in ALS pathogenesis. Cortical neuronal degeneration, apical dendritic regression, and dendritic spine loss along with enhanced glutamatergic excitation have been identified as pre-symptomatic features in the SOD1^G93A^ mouse cortical neurons, particularly Betz cells [105,106,107,108]. The Betz cells synapse extensively with cortical interneurons via their arborous dendritic processes, and are crucial in regulating cortical motor output [17]. The Betz cells possess long apical dendrite that extend towards the outer layers of the motor cortex and receive input from cortical interneurons [107]. Voltage–gated Kv1 channels (KCNA1) are extensively expressed on these apical dendrites and function as inward-rectifier currents preventing neuronal hyperexcitability [110]. Apical dendrite degeneration and dendritic spinal loss has been documented to be an early feature in the hSOD1^G93A^ mice [106,111,112], leading to cortical neuronal hyperexcitability.

More recently, pluripotent stem-cell derived motor neurons from ALS patients have been shown to exhibit hyperexcitability, and inhibition of this neuronal hyperexcitability appears to be neuroprotective [113]. Separately, upregulation of persistent Na^+^ currents has been reported in SOD-1 mouse cortical neurons, suggesting a direct link between cortical hyperexcitability and persistent Na^+^ conductances [104]. Inhibition of the persistent Na^+^ conductances by riluzole appears to be neuroprotective in cell models [104,114]. Importantly, upregulation of persistent Na^+^ conductances has also been well documented in ALS patients, accompanied by reduced K^+^ currents, and ultimately leading to axonal hyperexcitability, neurodegeneration, and development of clinical features of ALS [21,63,115,116].

A contrasting view has suggested that neuronal hyperexcitability may be neuroprotective in ALS, a notion only supported by some animal studies [117,118]. These findings contrasted with contemporary mouse model studies [104,105,106,107,108], and that antagonism of hyperexcitability was neuroprotective [113]. Separately, normalization of astrocyte dysfunction, an important regulator of glutamate-mediated excitotoxicity, by “knocking-down” the expression of mutated SOD-1 genes was also shown to be neuroprotective [119]. Furthermore, neuronal hyperexcitability was identified in transgenic mouse models overexpressing the human endogenous retrovirus-K genes [120]. Lastly, the clinical effectiveness of the anti-glutaminergic agent riluzole in ALS patients [57,58] lends additional credence for the pathogenic importance of cortical hyperexcitability in ALS.

## 5. Contrasting Views on ALS Pathogenesis and Site of Disease Onset

The site of disease onset (central versus peripheral) remains a matter of intense debate in the understating of ALS pathogenesis [3]. While primacy of the upper motor neurons in ALS pathogenesis was first proposed by Charcot [2], and later expounded in the dying forward hypothesis [7], with support provided by a clinical, neurophysiological, and molecular studies (see above), two other hypotheses have argued against the primacy of cortical dysfunction. The dying-back hypothesis proposed that ALS was primarily a disorder of lower motor neurons, with pathogens retrogradely transported from the neuromuscular junction to the cell body and central nervous system where they exert deleterious effects. Support for the dying back process has been predominantly provided by transgenic mouse models studies which have documented abnormalities of axonal transport as an early pathogenic feature predating the development of neuronal degeneration [8]. In addition, dysfunction of the distal axonal processes, preceding motor neuronal degeneration [121,122,123], have provided additional support for the dying-back hypothesis. A limitation of the dying-back hypothesis pertains to an absence of identifiable pathogens and that widespread cortical dysfunction documented in ALS patients remains difficult to reconcile with such a pathogenic process [124]. In addition, the absence of central pathology in ALS mimicking disorders such as Kennedy’s disease or poliomyelitis provides a further argument against a dying-back process [51,93].

The independent degeneration hypothesis proposed that upper and lower motor neurons degenerate independently and in a stochastic manner [125]. Neuropathological studies have provided evidence for such a process, documenting an absence of correlation between upper and lower motor neuronal dysfunction in ALS [126,127]. These correlative studies were potentially limited by anatomical and functional complexity of the corticomotoneuronal system [93,128]. More recently, clinical studies have provided additional evidence by documenting an absence of correlation between upper and lower motor neuron signs [10]. These clinical observations were not accompanied by neurophysiological assessments, and consequently subclinical upper or lower motor neuron dysfunction may have been undetected. In any event, future studies combining clinical, physiological, molecular and imaging techniques may help resolve this vital issue that could yet unlock therapeutic targets and treatment strategies in ALS.

## 6. Novel Neurophysiological Diagnostic Biomarkers for ALS

In the absence of a pathognomic diagnostic test, the diagnosis of ALS relies on identification of a concomitant upper and lower motor neuron dysfunction, with evidence of disease progression across different body regions [1,129]. The clinically based criteria (El-Escorial and Airlie House, also known as revised El-Escorial) were developed to facilitate an earlier and more definitive diagnosis of ALS, primarily for clinical drug trials. Unfortunately, the sensitivity of the clinically based criteria is limited, particularly in the early stages of ALS [124,130,131,132], leading to significant diagnostic delays and thereby a delay in institution of neuroprotective therapies and recruitment into therapeutic trials.

In order to increase the diagnostic yield, a neurophysiologically based Awaji-Shima criteria was proposed [129], whereby needle electromyography (EMG) findings of ongoing neurogenic changes (fibrillation potentials/positive sharp waves) and fasciculations were deemed equivalent to LMN signs. While the Awaji criteria exhibited greater sensitivity [130,131,132], the main limitation pertained to a reliance on clinical features for identifying upper motor neuron dysfunction [133].

The identification of cortical hyperexcitability as an early and specific feature of ALS suggested a potential utility as an objective diagnostic biomarker in ALS. Importantly, the presence of cortical hyperexcitability (as heralded by a significant reduction of SICI) reliably differentiated ALS from neuromuscular mimicking disorders [53,81,116], enhancing the diagnosis of ALS by eight months [13]. More recently, incorporation of reduced SICI and CSP duration in a novel ALS diagnostic index, reliably differentiated ALS from mimicking neuromuscular diseases at an early stage in the disease process [134], thereby providing class I evidence for a diagnostic utility of the threshold tracking TMS technique. Establishing an earlier and more definite diagnosis of ALS by using objective TMS biomarkers of cortical dysfunction, would enable an earlier recruitment of ALS patients into clinical trials during the putative therapeutic window period [15], where neuroprotective therapies are likely to be effective.

## Figures and Tables

**Figure 1 ijms-20-02818-f001:**
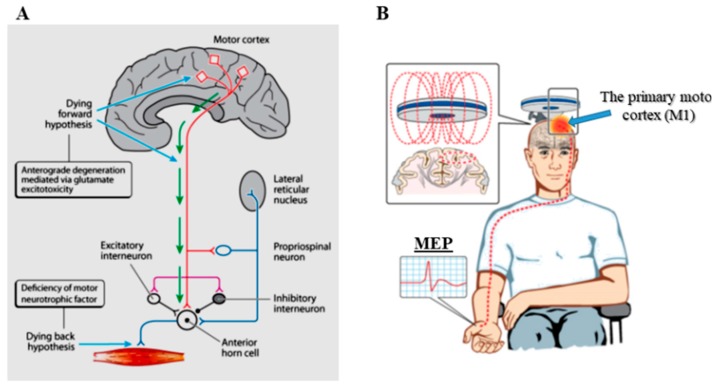
(**A**) The three theories for ALS onset are illustrated. The dying forward hypothesis proposes that ALS is primarily a disorder of the corticomotoneurons (red colour), which connect monosynaptically with the anterior horn cells, and mediate anterograde motor neuron degeneration via glutamate excitotoxicity. In contrast, the dying back hypothesis proposes that ALS begins within muscles or at the neuromuscular junction, with noxious factors transported retrogradely from the periphery to the axon cell body, where they exert toxic effects. The independent degeneration hypothesis proposed that upper and lower motor neuron degeneration occurs independently. (**B**) Transcranial magnetic stimulation excites neurons in the underlying motor cortex, with motor evoked potentials (MEP) recorded over the contralateral muscle. It is a sensitive measure of cortical excitability.

**Figure 2 ijms-20-02818-f002:**
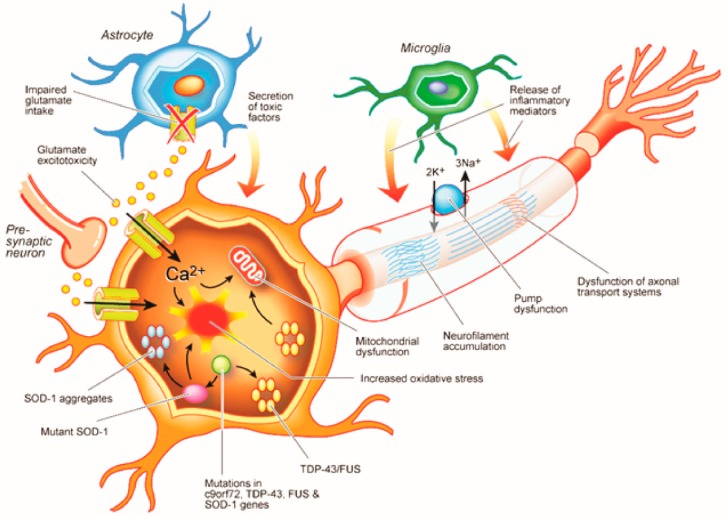
Amyotrophic lateral sclerosis (ALS) appears to be mediated by a complex interaction between molecular and genetic pathways. Reduced uptake of glutamate from the synaptic cleft, leading to glutamate excitotoxicity, is mediated by dysfunction of the astrocytic excitatory amino acid transporter 2 (EAAT2). The resulting glutamate-induced excitotoxicity induces neurodegeneration through activation of Ca^2+^-dependent enzymatic pathways. Mutations in the c9orf72, TDP-43 and fused in sarcoma (FUS) genes result in dysregulated RNA metabolism leading to abnormalities of translation and formation of intracellular neuronal aggregates. Mutations in the superoxide dismuates-1 (SOD-1) gene increases oxidative stress, induces mitochondrial dysfunction, leads to intracellular aggregates, and defective axonal transportation. Separately, microglia activation results in secretion of proinflammatory cytokines and neurotoxicity.

**Figure 3 ijms-20-02818-f003:**
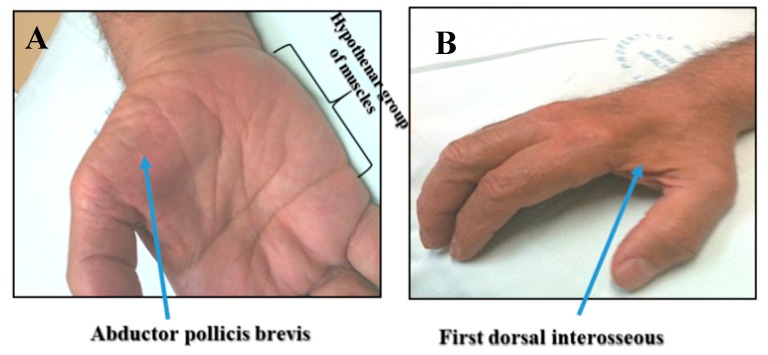
Dissociated muscle atrophy is a specific feature of ALS, characterized by preferential wasting of the (**A**) abductor pollicis brevis and (**B**) first dorsal interosseous muscles, when compared to the hypothenar group of muscles.

**Figure 4 ijms-20-02818-f004:**
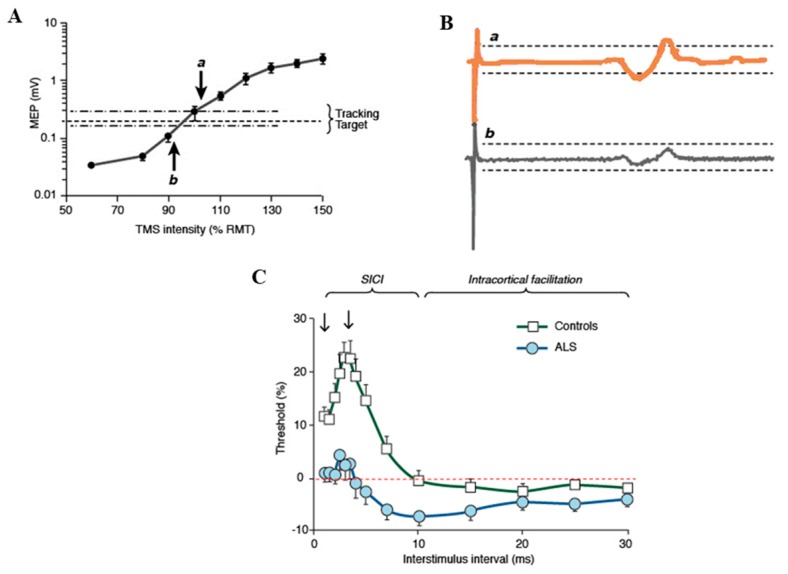
(**A**) A tracking target of 0.2 mV (± 20%), which lies in the steepest portion of the stimulus response curve, is tracked. As such, larger variations in the motor evoked potential (MEP) amplitude translate to smaller variations in transcranial magnetic stimulation (TMS) intensity. (**B**) When the MEP amplitude is larger than the tracking target (***a***) the intensity is reduced on subsequent stimulus, while when MEP is smaller (***b***) than the TMS, intensity is increased. (**C**) Short interval intracortical inhibition (SICI) is reflected by conditioned-test stimulus intensity being greater than zero (red dotted line), while the converse is true for intracortical facilitation (ICF). In amyotrophic lateral sclerosis (ALS), there is a significant reduction in SICI and an increase in ICF, indicative of cortical hyperexcitability.

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
