# Peer review of "Pathophysiology and Diagnosis of ALS: Insights from Advances in Neurophysiological Techniques"

_ijms, 2019, doi:10.3390/ijms20112818_

Reviewer 1 Report

Thank you for the kind invitation to review “Pathophysiology and diagnosis of ALS: Insights from advances in neurophysiological techniques” by van den Bos et al. This is a very interesting and topical review, which I feel would garner interest from the wider neuromuscular field. I have a few points for consideration, to improve the manuscript:

In the introduction the authors mention the three hypotheses from which the article is centred on. I feel that to make this more palatable and accessible to non-ALS specialists, the inclusion of a figure to emphasise and summarise these three concepts would help the overall narrative.

The inclusion of some basic background on TMS as a technique would help a non-specialist reader.

The authors discuss EAAT2, a brief line to clarify its biological role would help orientate the reader around its impact in terms of ALS.

The inclusion of a figure around some of the molecular aspects of the review would be welcomed.

Figure 1 is far too small – perhaps break up the composite image.

Line 51: reword the sentence beginning “The present review will an overview…” does not make sense in its current state.

Lines 56 – 62: This paragraph needs a re-write from scratch, its rather vague and long-winded the sentence structure and prose is lacking throughout. It its current state, it doesn’t help focus the reader on the topic – sentences are overly long, and bereft of any specific detail, more a generalisation – not accessible to a non-specialist.

Line 64 – 66: The first sentence of this paragraph is overly long and poorly constructed, please revise.

Line 72: Can the authors please clarify “…naturally occurring ALS animal models…”

Line 77- 80: The authors discuss preferential atrophy of certain muscle groups. Again, I feel a figure (perhaps clinical images) to emphasise this point would help the narrative.

Line 81: The authors describe dissociated pattern of atrophy specific to ALS; perhaps contrast this against other neuromuscular disease, as this appears to be a very key point.

Author Response

Comments to author

In the introduction the authors mention the three hypotheses from which the article is centred on. I feel that to make this more palatable and accessible to non-ALS specialists, the inclusion of a figure to emphasise and summarise these three concepts would help the overall narrative.

Response

A figure summarising the three concepts of disease onset have bene included.

Comments to author

The inclusion of some basic background on TMS as a technique would help a non-specialist reader.

Response

A discussion on the basic background on TMS is provided (Pages 6 and 7, lines 133-143).

Comments to author

The authors discuss EAAT2, a brief line to clarify its biological role would help orientate the reader around its impact in terms of ALS.

Response

The biological role of EAAT2 has been clarified (Page 12, lines 274-277).

Comments to author

The inclusion of a figure around some of the molecular aspects of the review would be welcomed.

Response

A new figure (figure 2) highlighting the molecular aspects of the review is included in the revised manuscript.  

Comments to author

Figure 1 is far too small – perhaps break up the composite image.

Response

The figure has been simplified, omitting the picture depicting the brain.  We felt that leaving the three figures together is important for impact.   

Comments to author

Line 51: reword the sentence beginning “The present review will an overview…” does not make sense in its current state.

Response

This sentence has been reworded for clarity (Page 4, lines 71-74).

Comments to author

Lines 56 – 62: This paragraph needs a re-write from scratch, its rather vague and long-winded the sentence structure and prose is lacking throughout. It its current state, it doesn’t help focus the reader on the topic – sentences are overly long, and bereft of any specific detail, more a generalisation – not accessible to a non-specialist.

Response

The paragraph has been re-written (Page 4, lines 77-87).

Comments to author

Line 64 – 66: The first sentence of this paragraph is overly long and poorly constructed, please revise.

Response

The sentence has been revised (Page 4, lines 90-92).

Comments to author

Line 72: Can the authors please clarify “…naturally occurring ALS animal models…”

Response

ALS is a neurodegenerative disorder that is unique to humans and does not appear in the animal world.  In addition, transgenic ALS models do not faithfully reproduce the human disease.  This point has been clarified in the revised manuscript (Page 5, lines 99-101). 

Comments to author

Line 77- 80: The authors discuss preferential atrophy of certain muscle groups. Again, I feel a figure (perhaps clinical images) to emphasise this point would help the narrative.

Response

A figure depicting dissociated muscle atrophy is included (figure 3).

Comments to author

Line 81: The authors describe dissociated pattern of atrophy specific to ALS; perhaps contrast this against other neuromuscular disease, as this appears to be a very key point.

Response

The dissociated pattern of muscle atrophy is specific for ALS when compared to other neuromuscular disorders.  This point is now highlighted in the revised manuscript (Page, 5, lines 110-114). 

Reviewer 2

Reviewer 2 Report

The review by Mehdi A.J. et. al. “Pathophysiology and diagnosis of ALS: Insights from

advances in neurophysiological techniques” highlights some important information in the context of the heterogeneity of the clinical phenotype and of the misdiagnosis of amyotrophic lateral sclerosis (ALS).

The authors have provided interesting evidence how transcranial magnetic stimulation (TMS) techniques will be useful to identify biomarkers involved in ALS pathogenesis.  The review is overall well written with regard to scientific content and does not need major revision. The references appear to be appropriate for the content.

However, the authors might consider editing and proof reading the content for grammar and sentence structure. Due to lack of clarity in various sentences across the text, the scientific meaning gets confounded.

the abstract would profit from a clearer reasoning with avoidance of redundancy with regard to the main text

line 33- point out that three different theories will be discussed

line 56- use abbreviation for ALS instead of full name (abbreviation had been explained previously)

line 58- explain what the multiple steps that precede ALS are

line 65- pathognomonic instead of pathognomic

line 109- expand the lists

line 181- move explanation about  GABA to the beginning of the text when it is first mentioned

Author Response

Comments to author

However, the authors might consider editing and proof reading the content for grammar and sentence structure. Due to lack of clarity in various sentences across the text, the scientific meaning gets confounded.

Response

We have thoroughly edited the manuscript as suggested by reviewer #2, incorporating the comments of both reviewers. 

Comments to author

The abstract would profit from a clearer reasoning with avoidance of redundancy with regard to the main text

Response

The abstract has been edited so as to make it clearer and redundancy has been avoided (Page 2, lines 2-37). 

Comments to author

Line 33- point out that three different theories will be discussed.

Response

This has been included in the revised manuscript (Page 3, line 51). 

Comments to author

Line 56- use abbreviation for ALS instead of full name (abbreviation had been explained previously)

Response

This sentence has been edited such that the term “Amyotrophic lateral sclerosis” is not the first term (Page 4, lines 77).

Comments to author

Line 58- explain what the multiple steps that precede ALS are.

Response

The “multi-step” process in ALS was based on modelling studies suggesting that six steps are required for onset of ALS. At present, we don’t understand the precise factors or steps involved, but postulate that it’s a combination of genetic, environmental and epigenetic factors. This has been clarified in the revised manuscript (Page 4, lines 81-87).    

Comments to author

 Line 65- pathognomonic instead of pathognomic.

Response

The typographical error has been corrected.

Comments to author

Line 109- expand the lists

Response

The list of paired-pulse TMS biomarkers is comprehensive.  While there are additional paired-pulse TMS biomarkers, such as long interval intracortical inhibition and short afferent inhibition, these parameters are rarely used in the understanding of ALS pathogenesis.  If the reviewer feels strongly about this topic we would be happy to include.  

Comments to author

Line 181- move explanation about GABA to the beginning of the text when it is first mentioned.

Response

This sentence is reflecting the physiological processes underlying CSP duration.  The GABAergic circuits mediating CSP duration are distinct to those medicating SICI.  As such we feel that this sentence should remain in the present position.